# Current Epidemiological Characteristics of Imported Malaria, Vector Control Status and Malaria Elimination Prospects in the Gulf Cooperation Council (GCC) Countries

**DOI:** 10.3390/microorganisms9071431

**Published:** 2021-07-02

**Authors:** Jamshaid Iqbal, Suhail Ahmad, Ali Sher, Mohammad Al-Awadhi

**Affiliations:** 1Department of Microbiology, Faculty of Medicine, Kuwait University, Jabriya 46300, Kuwait; suhail.ahmad@ku.edu.kw (S.A.); paramo1983@hotmail.com (M.A.-A.); 2Ministry of Health, Infectious Diseases Hospital, Shuwaikh 70030, Kuwait; alisher02@gmail.com

**Keywords:** GCC countries, imported malaria, expatriates, epidemiology, prevalence

## Abstract

Malaria is the most common vector-borne parasitic infection causing significant human morbidity and mortality in nearly 90 tropical/sub-tropical countries worldwide. Significant differences exist in the incidence of malaria cases, dominant *Plasmodium* species, drug-resistant strains and mortality rates in different countries. Six Gulf Cooperation Council (GCC) countries (Bahrain, Kuwait, Qatar, Oman, Saudi Arabia and United Arab Emirates, UAE) in the Middle East region with similar climates, population demographics and economic prosperity are aiming to achieve malaria elimination. In this narrative review, all studies indexed in PubMed describing epidemiological characteristics of indigenous and imported malaria cases, vector control status and how malaria infections can be controlled to achieve malaria elimination in GCC countries were reviewed and discussed. These studies have shown that indigenous malaria cases are absent in Bahrain, Kuwait, Qatar and UAE and have progressively declined in Oman and Saudi Arabia. However, imported malaria cases continue to occur as GCC countries have large expatriate populations originating from malaria-endemic countries. Various malaria control and prevention strategies adopted by GCC countries including more stringent measures to reduce the likelihood of importing malaria cases by prior screening of newly arriving expatriates and vector elimination programs are likely to lead to malaria elimination in this region.

## 1. Introduction

Significant progress has been made in human healthcare during the last several decades. However, infectious diseases caused by human parasites continue to inflict considerable suffering and death worldwide, particularly in poor and developing countries. Estimates of global burden of diseases have suggested that more than 2 billion cases of parasitic infections afflicted mankind in the year 2013 [1]. Among human parasitic diseases, vector-borne parasitic infections are particularly important as they are responsible for more than 15% of all infectious diseases worldwide, causing more than 600,000 deaths every year, mostly in poor tropical/sub-tropical countries [2,3]. Malaria is an important vector-borne parasitic infection which alone is responsible for more than 400,000 fatalities every year around the world with most of the deaths occurring in younger (less than 5-year old) children [3].

Five *Plasmodium* species, *P. falciparum, P. vivax, P. ovale, P. malariae* and *P. knowlesi*, cause malaria in humans and the disease spreads from person to person through the mosquito vector. Zoonotic malaria cases have also been reported due to *P. knowlesi* in some countries in Southeast Asia. Malaria cases are reported from 87 different countries and significant differences exist in the incidence of malaria cases, dominant *Plasmodium* species causing the infections, resistance to antimalarial drugs and mortality rates among infected patients in different countries. Infections by *P. falciparum* and *P. vivax* are most common and account for most of the global malaria cases. Most malaria infections in Africa are caused by *P. falciparum* while the highest burden of *P. vivax* infections is seen in South/Southeast Asian countries (including India) and South America [4]. In most African countries, *P. vivax* infection is absent or very rare due to lack of Duffy antigen receptor for chemokines on the surface of red blood cells that is involved in the parasite invasion of erythrocytes among the population. However, since 2010, there has been a growing evidence of presence of *P. vivax* infections in all regions of Africa, including *P. vivax*-infected Duffy-negative individuals [5]. Although *P. falciparum* is the more virulent and causes the majority of malarial mortality, particularly in Africa, increasing prevalence of *P. vivax* malaria in some countries, particularly in the Indian subcontinent, poses unique diagnostic and therapeutic challenges. Several reports have described acute respiratory distress syndrome, cerebral malaria, multi-organ failure, dyserythropoiesis and anemia following *P. vivax* infection [4,5]. The *P. vivax* malaria also triggers higher inflammatory responses and exacerbated clinical symptoms (fever and chills) than *P. falciparum* malaria [5]. The *P. vivax* infection also results in persistence of the parasite, as dormant liver-stage hypnozoites can cause recurrent episodes of malaria [4].

The United Nations Millennium Summit included three major infectious diseases, viz. malaria, human immunodeficiency virus (HIV) infection and tuberculosis for special attention and created a global fund to support control and eradication of these infections [6]. For malaria, the term ‘elimination’ is used when disease transmission is no longer occurring in a specific geographic area while the term ‘eradication’ is used to describe malaria elimination from the whole world. Concerted worldwide efforts during the first decade of the new millennium resulted in significant progress as nearly 1.5 billion malaria infections and 7.6 million malaria-related deaths were averted [6], according to the Malaria Atlas Project 2019, https://malariaatlas.org/malaria-burden-data-download/ accessed on 18 March 2021.

An important factor that has halted the progress against malaria in recent years is the emergence of drug-resistant strains of *Plasmodium* species which are currently posing serious threats to global malaria control efforts. Currently, the World Health Organization (WHO) recommends artemisinin-based combination therapies (ACT) as a first-line therapy in most malaria-endemic countries, as the older antimalarial drugs are now ineffective in treating malaria cases worldwide [7,8]. In addition, judicious use of appropriate chemical insecticides through long-lasting insecticidal nets (LLINs), insecticide-treated mosquito nets (ITNs) and indoor residual spraying (IRS) have also been useful [6]. Vector control through LLINs and IRS has reduced global malaria burden by nearly 40% in some countries during the last decade [7]. However, recent reports of the rapid spread of pyrethroid resistance in malaria vectors across Africa have documented the reduced effectiveness of LLINs, thus leading to increased malaria transmission in sub-Saharan Africa [7,9,10]. New types of LLINs (e.g., PBO-Py LLINs, Olyst Plus) are being manufactured which have shown improved effectiveness against pyrethroid-resistant mosquitoes [11].

A new paradigm shift that has occurred in many malaria non-endemic countries/regions, particularly among those located in the temperate climate of Europe and North America and a few oil-rich Arabian Gulf countries, is the emergence of imported malaria cases [12,13,14,15]. Many countries which were malaria free until a few years ago have seen a sharp increase in the number of imported malaria cases in recent years [12,13,14,15]. This increase has occurred as a result of increased international travel for business or tourism particularly from or to malaria-endemic countries, migratory movements for employment and migration of large number of refugees from war-torn countries into non-endemic/malaria-free countries [12,16,17,18].

Some other areas of concern have also been noted in recent years. Deletions in the *P. falciparum* histidine-rich protein (pfhrp)2 and pfhrp3 genes have been confirmed in 11 African countries [6,19,20]. Deletions of pfhrp2 and pfhrp3 genes reduce the diagnostic utility of rapid tests based on detection of HRP2. Mutations in PfKelch13 have also been detected in malaria patients in many malaria-endemic countries. Some of these mutations confer partial resistance to artemisinin, the first-line treatment for *P. falciparum* infections [4,21]. Of the 81 malaria-endemic countries that provided data for vector resistance to insecticides during 2010 to 2019, 28 countries detected resistance to all four of the most commonly used insecticide classes in at least one malaria vector and one collection site, and 73 detected resistances to at least one insecticide class. Only eight countries did not detect resistance to any insecticide class [6].

The effects of the COVID-19 pandemic on the substantial progress made in reducing the burden of malaria is still being evaluated. However, it is most likely that this pandemic will severely disrupt important malaria control interventions, i.e., distribution of LLINs, ITNs and antimalarial. It has been suggested that these disruptions will potentially lead to greater increases in malaria incidence in subsequent years and the malaria mortality will likely double from 386,400 to 768,600 individuals. In Nigeria alone, reducing case management for 6 months and delaying LLIN campaigns could result in 81,000 (44,000–119,000) additional deaths [6,22,23]. On the other hand, strict travel restrictions due to COVID-19 may have decreased the number of imported malaria cases in non-endemic countries. For instance, the number of imported malaria cases in Kuwait during the COVID-19 pandemic year 2020 decreased by more than 80%, as the State of Kuwait put in place strict travel restrictions on people coming to Kuwait from Africa and South Asia during this period (personal communication, Infectious Diseases Hospital, Kuwait).

We recently reviewed the epidemiology of malaria in Middle Eastern countries [12]. In this narrative review, all studies indexed in PubMed describing epidemiological characteristics of indigenous and imported malaria cases, status of malaria vectors and how these infections can be controlled to achieve malaria elimination in the six (Bahrain, Kuwait, Qatar, Oman, Saudi Arabia and United Arab Emirates, UAE) Gulf Cooperation Council (GCC) countries were reviewed and discussed.

## 2. Status of Malaria Infection and Vector Control in Gulf Cooperation Council (GCC) Countries

### 2.1. Status of Vector Control in GCC Countries

Six (Bahrain, Kuwait, Oman, Qatar, Saudi Arabia and United Arab Emirates) countries located in the Arabian Gulf region are collectively grouped as Gulf Cooperation Council (GCC) countries. These countries have many common dynamics such as a very hot and dry climate, high per capita income based on oil-based economy, modern health infrastructure, and a large and dynamic expatriate workforce. Successful control programs have disrupted local malaria transmission (indigenous malaria) in nearly all GCC countries except a few sites in Saudi Arabia and sporadic malaria cases in Oman, primarily due to the presence of the malaria vector [24,25]. However, all GCC countries report imported malaria cases among the incoming expatriates coming from malaria-endemic countries, especially from South Asia and Africa. Human malarial parasites are transmitted by the *Anopheles* mosquitoes, which has approximately 70 species capable of transmitting the infection. Of these, more than 41 are dominant vector species, and thus are of major concern to public health [26,27].

The major mosquito species that have been recognized to breed within GCC countries and can potentially act as vectors for indigenous spreading of malaria are listed in Table 1. While only one or few mosquito species have been identified to breed in Oman and Qatar, the largest number (*n* = 7) of *Anopheles* species have been detected to breed in Saudi Arabia, particularly in the southwestern region of the country. On the contrary, no mosquito species are known to breed in Bahrain, Kuwait and the United Arab Emirates.

### 2.2. Status of Malaria Infection in GCC Countries

The malaria-free status of GCC countries officially designated by the WHO is presented in Table 1. The incidence of indigenous malaria cases among the GCC countries reported during the period 2000–2019 are shown in Figure 1.

The data were adopted from WHO, World Malaria Reports, 2019 and 2020 and the Malaria Atlas Project 2019 (https://malariaatlas.org/malaria-burden-data-download/). Only two countries, Oman and Saudi Arabia, have reported indigenous malaria cases during the last two decades. However, the large expatriate populations in GCC countries, which mostly come from South/Southeast Asian (India, Pakistan, Nepal, Bangladesh, Sri Lanka, Philippines etc.) and African countries (Nigeria, Ethiopia, Sudan, Egypt etc.), pose a risk of importing infectious diseases such as malaria, tuberculosis and other parasitic infections that are endemic in their native countries [6,14,37,38,39,40,41,42,43]. Due to these reasons, imported cases of malaria continue to occur in GCC countries, especially among returning expatriate workers from Africa and the Indian subcontinent after their holidays or new expatriates who come to work in these countries. The number of imported malaria cases reported in recent studies from the GCC countries is shown in Figure 2. The indigenous and imported malaria cases reported in different recent studies from GCC countries are presented in Table 2. The data show that the number of imported malaria cases in different studies and the *Plasmodium* species causing malaria infections vary considerably even from the same location due to the large and highly dynamic expatriate population in the GCC countries. In addition, the data in Table 2 also highlight a gradual decrease in the number of indigenous cases (Oman and Saudi Arabia) over the years, as a consequence of malaria control/preventive strategies adopted by the countries.

It is expected that there will be a sharp decrease in the number of tropical infections including imported malaria cases in GCC countries in year 2020 and beyond due to prolonged travel restrictions imposed during the COVID-19 pandemic on individuals coming from Southeast Asian countries, Africa and the Indian subcontinent which contribute the bulk of imported malaria in most of the GCC countries. Various malaria control and prevention strategies have been adopted in almost all the GCC countries to prevent the influx of imported malaria. A brief account of the recent developments and the current epidemiology of imported malaria in GCC countries is described below.

#### 2.2.1. Status of Malaria and Vector Control in Bahrain

Bahrain is a small island nation on the eastern side of the Arabian Peninsula and has the smallest (~1.72 million) population among GCC countries. Approximately 50% of the population of Bahrain included expatriate workers or their dependents according to the 2010 census. According to the Ministry of Health, Bahrain Government, malaria was eradicated in Bahrain in 1979 but the WHO declared it malaria-free in 2012 [24,28]. The malaria-free status achieved by Bahrain was largely the result of vigorous control and surveillance measures initiated in 1946, which included insecticide spraying of the vector breeding sites and active case detection to prevent transmission and re-emergence of indigenous malaria transmission. In recent years, no malaria vector has been detected in Bahrain (Table 1). However, imported malaria cases continue to occur mostly due to the large expatriate population that is employed as an expatriate workforce and comes from malaria-endemic countries of Asia and Africa. Only one study has reported on the situation of malaria in Bahrain in the new millennium. The study reported 1572 cases of malaria that occurred during 1992 to 2001 and all were imported malaria cases. The study also showed a consistent decline in the number of annual malaria cases [29]. The malaria cases mostly occurred among expatriates originating from malaria-endemic countries. The study further showed that *P. vivax* was responsible for 86% (1346/1572) of all malaria cases, while only 220 (14%) cases were caused by *P. falciparum* (Table 2). Although this study also detected few mosquito breeding sites where *Anopheles* mosquito larvae could be detected, local transmission of infection was not reported [29]. In 2017, a total of 133 confirmed malaria cases were detected, of which 100% were imported, mainly from East Africa and the Indian subcontinent [37].

#### 2.2.2. Status of Imported Malaria in Kuwait

According to the Public Authority for Civil Information (PACI) the total population of ~4.5 million inhabitants in Kuwait in 2019 included only ~30% Kuwaiti nationals while the remaining 70% individuals comprised expatriates [52]. Most expatriates in Kuwait come from the Indian subcontinent, Southeast Asia and African countries that are endemic for malaria, tuberculosis, taeniasis and other infectious diseases [39,40,41,42,44,45]. Kuwait was the first country among the GCC countries which was declared malaria-free by the WHO in 1979 [24]. In Kuwait, malaria infection is a notifiable disease. Previous studies have shown that there is no indigenous transmission of malaria in Kuwait as the *Anopheles* mosquito vector is not found due to lack of freshwater bodies and extremely hot weather conditions during the summer months. All migrant workers arriving from malaria-endemic countries are screened for malaria infection at Health Centers before work permits are issued. Despite these measures, imported malaria cases have been detected regularly in multiple studies during the past 30 years among expatriate workers and their family members arriving from malaria-endemic countries of the Indian subcontinent, Southeast Asia and Africa [44,45,53]. This is like the problem of imported malaria in other neighboring GCC countries and other adjoining countries in the Middle East Region [32,43]. A brief account of the various epidemiological studies carried out recently in Kuwait is described below.

One study was carried out during 1985–2000 and reported an infection rate of more than 1200 malaria cases occurring every year [44]. As expected from the absence of the mosquito vector in Kuwait, all infections represented imported malaria cases. The patients included both newly arriving expatriate workers from endemic countries as well as expatriates who were already residing in Kuwait but had travelled to their respective countries and were returning to Kuwait after holidays in their native countries [44]. An earlier study also showed that 71% of all malaria cases were caused by *P. vivax* while the remaining 27% cases were due to *P. falciparum* in Kuwait [53]. Interestingly, these studies and the data obtained from a more recent epidemiological study have shown that the incidence of imported malaria cases in Kuwait have been declining steadily over the years, primarily due to proactive malaria control programs run by the Ministry of Health, Kuwait [44,45,53]. Furthermore, 80% of malaria cases were detected among young expatriate male adults aged 21–40 years. To control the influx of imported malaria cases in Kuwait, expatriate workers from the malaria-endemic countries are now required to have a malaria-free certificate before entering Kuwait [44,45].

The data reported in the most recent study have also shown that the highest number of imported malaria cases (1012 of 1913, 52.9%) were detected among expatriates originating from India, which is not surprising since Indians also form the largest single ethnic group among expatriates in Kuwait [45]. The study further showed that other major malaria-positive subjects originated from Pakistan (380 subjects), Afghanistan (94 subjects), Cameroon (54 cases), Sudan (48 cases) and Ghana (46 cases). Subjects from other African countries (165 cases) and some other countries (86 cases) accounted for the remaining cases among expatriates. As expected, only 18 Kuwaiti citizens tested positive for malaria infection and all these patients had travelled to malaria-endemic African countries. In contrast to an earlier study [44] in which 71% of all malaria cases were caused by *P. vivax*, most (1383 of 1913, 72.3%) malaria cases in the recent study were due to mixed *P. falciparum* and *P. vivax* infections while only small number of cases were caused by *P. falciparum* alone (361 cases) or *P. vivax* alone (124 cases) or *P. falciparum* and *P. ovale* (25 cases) or by *P. falciparum* and *P. malariae* (12 cases) mixed infections (Table 2) [45].

Another interesting observation of the latest study was that almost all mixed *P. falciparum* and *P. vivax* (1303 of 1383, 97.0%) infections occurred among malaria patients originating from India. The study also highlighted that certain risk factors for local transmission of infection have recently emerged even though no indigenous transmission of malaria cases has been detected in Kuwait. Recent data show that ecological and climate conditions are slowly changing in Kuwait due to increased planting to make Kuwait green. A genuine concern in this regard is that these altered ecological conditions may raise larval density levels of *A. stephensi* and *A. pulcherrimus* which may support local transmission of malaria infection. The study concluded that many challenges are needed to be overcome to prevent reintroduction and local transmission of malaria into malaria-free Kuwait [45].

Interestingly, the number of imported malaria cases in Kuwait during the COVID-19 pandemic year of 2020 have decreased by more than 80%, as the State of Kuwait put in place strict travel restrictions on people coming to Kuwait from Africa and South Asia during this period (Jamshaid Iqbal, unpublished data).

#### 2.2.3. Status of Malaria and Vector Control in Oman

The total population of Oman included 4.5 million inhabitants in 2020. According to the Omani Government statistics, Omani nationals comprised 57.7% of the total population while the remaining 42.3% individuals were expatriates who had come to work in Oman or their dependents [54]. Oman was a malaria-endemic country till the 1960s as the climate of Oman is suitable for the mosquito vectors which were responsible for the indigenous transmission of infection, and nearly 30% of the population had malaria, especially in the northern areas of the country, mostly with *P. falciparum* infection [30,31]. A malaria control program was initiated in 1975 with several key objectives. These included setting up of malaria training centers for local staff, identification of the source (vector, *An. culicifacies*) and its reduction by larvivorous fish *Aphanius dispar*, widespread use of larvicides and continuous tracking of larval control and weekly chloroquine prophylaxis treatments of school children in certain highly endemic areas during 1973–1979 [30]. Despite these measures, Oman was still considered an endemic country until 1990 as nearly 33,000 malaria cases were reported every year.

In 1991, the Omani Ministry of Health introduced a malaria elimination program with a focus on completely interrupting malaria transmission with a strong surveillance system in place by eliminating the existing reservoir of infected cases, screening of all travelers from East African countries and prompt active case detection and management. These malaria control programs actively pursued by the Ministry of Health, Oman resulted in successfully reducing the incidence of malaria to nearly 1 case per 10,000 people by the year 2000 [31,32]. The data reported by the Ministry of Health further showed that most cases in 2004 were caused by *P. vivax* and no locally acquired cases were reported between 2004 and 2006. Oman was considered as a ‘hugely successful program’ and was officially recognized by the WHO as a center of excellence for malaria microscopy in the Eastern Mediterranean Region and training programs for the neighboring countries in the region until 2014. However, according to the Ministry of Health Annual Reports, a small number of imported cases and indigenous malaria cases have occurred in recent years (Table 1 and Figure 2). The control efforts were further intensified in 2016 by strengthening malaria surveillance as a core intervention for better monitoring and evaluating the intervention, as outlined in the WHO’s Global Technical Strategy [13]. Currently, only a few isolated outbreaks of malaria have been reported.

Only limited recently published data are available regarding malaria cases in Oman. Local transmission of *P. vivax* infections has been registered in small outbreaks since 2007 due to continued presence of the vector *A. culicifacies* [31,32]. The last documented indigenous malaria case was detected in 2011 [6]. However, a local outbreak of *P. vivax* involving 54 cases over a span of 50 days occurred in 2014 among migrant workers [31]. Fingerprinting studies based on DNA sequence data obtained for circumsporozoite protein (csp), merozoite surface protein 1 (msp1) and merozoite surface protein 3 (msp3) found low genetic diversity and a high genetic distance with most of the reference strains coming from endemic countries. Although parasite larvae were isolated from irrigation pools and large water tanks used in construction sites, this small outbreak of imported malaria cases was not associated with re-emergence of malaria transmission in Oman, as no new cases were reported after the outbreak had ended [31]. However, according to the Ministry of Health Annual Reports, a small number of indigenous malaria cases have occurred in recent years. A total of 15 indigenous cases were reported in 2019 in addition to 1323 imported malaria cases by the Ministry of Health in their Annual Report, of which 1080 cases were due to *P. falciparum*, 206 cases due to *P. vivax* and 52 due to other species [32]. The data are also presented in Table 2.

#### 2.2.4. Status of Imported Malaria in Qatar

Qatar’s population of 2.6 million inhabitants in 2017 included 12% Qatari nationals and 88% expatriates. Malaria transmission was eliminated in Qatar in the 1970s, but the country was declared malaria-free by WHO in 2012 [24]. *An. stephensi* and *An. multicolor* were the predominant malaria vectors in Qatar, which were eliminated by an intense vector control/elimination program [33]. Several studies have investigated the current situation of malaria, particularly imported malaria in the country in the last few decades. These studies have shown that all malaria cases are travel-related and occur in expatriates who have previously travelled to their native, malaria-endemic countries or among Qatari nationals who had travelled to malaria-endemic countries. One large study determined the incidence of malaria over a period of several years and reported that the incidence decreased from 58.6 cases/100,000 population in 1997 to 9.5 cases/100,000 population in 2004, but this declining trend was reversed from 2005 onward. All patients were imported malaria cases and the increasing incidence of malaria in the later part of the study was related to increased influx of expatriate workers and increased travel of expatriates to their native, malaria-endemic countries [33]. A subsequent study reported similar findings as a total of 4092 malaria cases were reported during 2008–2015 with the majority of cases occurring among young non-Qatari males, particularly during the months of July, August and September which coincide with the return of most expatriates after annual holidays in their native countries, of which 812 imported cases were detected in individuals from India and 772 cases were from Pakistan (Table 2) [46]. This study further showed that *P. vivax* was the main etiologic agent, accounting for 79% of imported cases.

Few recent studies have also reported on the current situation of malaria in Qatar. One study using molecular diagnostic methods showed that most imported malaria cases in Qatar between 2013 to 2016 were due to *P. vivax*, while *P. falciparum* and *P. falciparum*/*P. vivax* mixed infections occurred with much less frequency [47]. The origin of *P. vivax* infections was the Indian subcontinent, while *P. falciparum* was mostly detected among malaria cases from individuals returning from Africa. The study further showed that *P. falciparum* strains were genetically highly variable, and a high prevalence of mutations in genes implicated in drug resistance, including Pfkelch13, was also observed among malaria parasites. Although common mutations in Pfkelch13 which confer resistance to artemisinin were not identified, several novel nonsynonymous mutations were detected. However, their role in conferring resistance of malarial parasite to artemisinin was not established [47]. High genetic diversity was also seen among imported *P. vivax* parasites in another recent study from Qatar [55]. This study showed that the *Plasmodium* species strains from patients of Indian origin were genotypically comparable to parasites from India and strains from patients of African origin were genotypically comparable to parasites from Sudan and Ethiopia. The study also detected gametocytes by highly sensitive quantitative real-time PCR-based methods. Though all cases represented imported malaria, detection of gametocytes suggested the possibility of re-introduction of malaria in Qatar. The authors concluded that influx of genetically diverse *P. falciparum*, with multiple drug resistance markers, represents a threat for the reestablishment of drug-resistant malaria into Qatar and other GCC countries. The study also highlighted the impact of mass international migration on the reintroduction of malaria into malaria-free countries with none or limited local transmission. Abdien et al. [56] screened blood donors in another recent study and also reported an increasing trend in the number of malaria cases in Qatar from 2013 to 2017, with an overall positivity rate of 0.2% among blood donations.

#### 2.2.5. Status of Malaria and Vector Control in Saudi Arabia

Saudi Arabia is the largest and is also the most populous (population of ~35 million inhabitants in 2020) country among the GCC countries as well as in the Arabian Peninsula. Similar to other GCC countries, Saudi Arabia is also home to many expatriates and their dependents, mostly originating from malaria-endemic countries. According to the Kingdom of Saudi Arabia Government statistics, Saudis accounted for 63.1% of the total population while expatriates comprised the remaining 31.8% of the individuals residing in Saudi Arabia in 2016 [57]. Malaria transmission in Saudi Arabia is considered low and unstable [34,35,58,59,60]. Although the number of indigenous malaria cases reported from Saudi Arabia have been declining in recent years, Saudi Arabia is the only GCC country from which both imported and locally acquired malaria cases have been regularly reported [6,13,48,49,50,51]. The indigenous malaria cases have mainly been reported from the Jazan and Aseer provinces located in the southwestern region of the country bordering Yemen, another malaria-endemic country in the Arabian Peninsula which is not included among GCC countries [48,50,59,60]. According to the latest WHO global malaria report, 38 cases of locally transmitted malaria cases were reported from Saudi Arabia in 2019 [6].

Saudi Arabia was endemic for malaria before the start of the new millennium. However, much progress has been made in the fight against indigenous malaria cases in Saudi Arabia in the past two decades. The country experienced a major outbreak of malaria infections in 1998, however, the number of indigenous malaria cases have declined sharply by 99.8% between 1998 and 2012 due to intensive malaria control programs initiated by the government of Saudi Arabia [38,48,58]. This sharp decline in the number of malaria cases was achieved mainly by rapid scaling up of vector control measures, adoption of artesunate and sulfadoxine/pyrimethamine combination treatment and a regional partnership initiated in 2007 for a malaria-free Arabian Peninsula [58]. The main aim of the draft Global Technical Strategy adopted in 2002 was to eliminate malaria from at least 10 countries by 2020 and Saudi Arabia and Yemen in the Arabian Peninsula are two countries yet to achieve malaria elimination [48,58]. The action plan to eliminate malaria from Saudi Arabia was revised in 2004 to include prevention of re-introduction of malaria into regions which had previously been declared malaria free. This was planned to be achieved by removing foci of active transmission in the two (Makkah and Medina areas) holy places frequently visited by Muslim pilgrims by increased surveillance and control of mosquito breeding foci and by eliminating malaria from the endemic Jazan and Aseer provinces [48,50,58,59,60].

Multiple studies have described the situation of indigenous and imported malaria cases in Saudi Arabia in recent years. A study carried out in Jazan province that analyzed data on malaria cases between the year 2000 and 2014 reported a declining trend in indigenous malaria cases with an average annual incidence of 0.03 cases per 1000 population [48]. In this study, 15458 total malaria cases were reported from this region of which 9930 (64%) cases were due to imported malaria while 5522 (36%) cases were indigenous cases acquired locally. Interestingly, the number of indigenous cases progressively declined from a high of 2756 cases in 2000 to only 15 cases in 2014. On the contrary, the imported malaria cases generally increased from a low of 250 in 2000 to 830 in 2014 but also showed large variations during intervening years with 1705 and 1310 imported cases detected in 2007 and 2009, respectively [48]. The data are also presented in Table 2. Although a similar but sometimes inconsistent decline for indigenous malaria cases was also seen in the Aseer region (from a high of 511 cases in 2000 to a low of 51 cases in 2015), the number of imported malaria cases showed a declining trend during 2000 to 2010 but an upward trend during 2011 to 2015 [59]. The identification of *Plasmodium* spp. by molecular methods in another study described the dominance of *P. falciparum* in Saudi Arabia [61]. The study showed that more than 98% of 371 malaria-positive blood specimens were due to *P. falciparum*, while only 7 cases had *P. falciparum/P. vivax* mixed infection. Surprisingly, no *P. vivax* monoinfection was detected in this study.

Alshahrani et al. surveyed 1840 individuals from 54 villages in the Province of Aseer during the years 2006–2007 using passive case detection survey. The authors reported that 49 (2.7%) individuals had malaria; only one individual had *P. vivax* infection while the remaining 48 individuals had *P. falciparum* infection [59]. Furthermore, 18 (36.7%) malaria cases were due to indigenous transmission of infection while the remaining 31 were imported cases from malaria-endemic countries [59]. The data are also presented in Table 2. The situation of malaria from Jazan, a low malaria transmission area in southwestern Saudi Arabia, was also studied more recently by Hawash et al. [35]. The authors detected malaria in 30 patients from August 2016 to September 2018 using a species-specific nested PCR assay. The study showed that 80% of malaria cases were imported, 76.6% of patients were infected with *P. falciparum*, 16.6% subjects presented with *P. vivax* infection and only 6.6% of the subjects presented with mixed *P. falciparum/P. vivax* infection [35]. In another study, Memish et al. [49] examined malaria patients and the parasite type in Makkah region which hosts millions of immigrants yearly during Hajj to monitor the incidence of malaria between 2008 and 2011. The authors detected 318 malaria cases, mostly (95%) among non-Saudis. Malaria-positive patients mostly (62%) originated from Pakistan, Nigeria and India. The study also showed that malaria infections were mostly caused by *P. falciparum* (67%) followed by *P. vivax* (32%) [49]. Several *Anopheles* species have been detected to breed in Saudi Arabia, particularly in the southwestern region of the country, with continued low and unstable indigenous transmission of malaria infection [34,35].

Soliman et al. [51] and Dafalla et al. [62] investigated the presence of different gene mutations concerning antimalarial drug resistance (pfdhfr, pfdhps, pfmdr1, pfcrt, pfcytb, pfkelch13) to identify whether drug-resistant alleles are present in Saudi Arabia. Among 26 positive cases, 13 were infected with *P. falciparum* (including 4 cases due to indigenous transmission) and 13 with *P. vivax*. No mutation in genes related to resistance to artemisinin or atovaquone was detected. The chloroquine resistance alleles were detected in 31% of samples and point mutations in the pfdhfr and pfdhps genes involved in antifolate drug resistance were also detected [62]. Although several mutations in the k13-propeller gene were detected, no widespread artemisinin-resistant polymorphisms were reported. Another study carried out in the Jazan region, southwestern Saudi Arabia, where artesunate plus sulfadoxine/pyrimethamine has been used since 2007 as a first-line treatment for uncomplicated *P. falciparum* malaria, investigated the prevalence of mutations associated with artemisinin and sulfadoxine/pyrimethamine resistance in the parasites circulating in this region. Polymorphisms in the propeller domain of the *P. falciparum* k13 (pfkelch13) gene and point mutations in the pfdhfr and pfdhps genes were identified by DNA sequencing [63]. Although no mutations were found in any of the 151 isolates in the pfkelch13 propeller domain, point mutations in pfdhfr and pfdhps genes were detected in 90.7% (137/151) of the isolates including double (N51I + S108N) and triple (N51I + C59R + S108N) pfdhfr mutations. Furthermore, the mutation A437G and double mutations A437G + K540E were also found in pfdhps gene. The study identified a high prevalence of point mutations in the pfdhfr and pfdhps genes including quintuple and quadruple mutant pfdhfr-pfdhps genotypes for the first time in the Jazan region of Saudi Arabia and the Arabian Peninsula. Although drug resistance-conferring pfkelch13 mutations were not found, mutations in the pfdhfr and pfdhps genes were detected. The authors concluded that their observations undermine the efficacy of artemisinin-based combination therapy in the Jazan region in Saudi Arabia [63].

Saudi Arabia needs to strengthen malaria surveillance as a core intervention, especially vector control operations in malaria foci, among high-risk populations such as undocumented migrants from neighboring Yemen, as well as communities living along the border with Yemen, to achieve malaria-free status. In addition, to control the influx of imported malaria cases in Saudi Arabia, expatriate workers from the malaria-endemic regions/areas should be required to have a malaria-free certificate before entering the country, as was effectively implemented in Kuwait [44,45].

#### 2.2.6. Status of Imported Malaria in the United Arab Emirates (UAE)

The UAE is currently the economic hub in the Middle East as it includes Dubai which is the most vibrant city in the region. The total population of nearly 10 million individuals in UAE in 2020 included only 20% Emirati nationals while the remaining 80% inhabitants were expatriates [64]. Only limited information is available on the incidence of malaria infections in UAE in recent years. According to the latest world malaria report, UAE is certified by the WHO to be free of indigenous transmission of malaria since 2007 [24]. However, imported malaria cases continue to be reported every year, mostly among migrant workers or returning travelers from malaria-endemic countries [2,3]. Only one recent study has reported on the epidemiological and clinical characteristics of imported malaria cases in UAE. This study, carried out at a large hospital in Dubai during 2008 to 2010, found 629 malaria cases which required 162 hospitalizations, including 8 patients who also required intensive care support. The study also reported one malaria-related death. The study found that most (493 of 629, 78%) of the malaria cases were due to *P. vivax* infection while only 122 (19%) cases were caused by *P. falciparum*. The study also reported 14 cases of mixed *P. vivax/P. falciparum* infections [36]. The data also showed that most (90.1%) of the infected individuals were expatriates from India or Pakistan, as shown in Table 2. Both India and Pakistan are malaria-endemic countries and their nationals are also the most common ethnic groups among the expatriate population in UAE. Nearly 7% of the malaria patients were from sub-Saharan Africa [36]. Surprisingly, no malaria cases were detected among the local Emirati population. Thus, imported malaria remains an important cause of morbidity in the UAE, like other GCC countries.

## 3. Malaria Status among Countries Contributing Most of the Imported Malaria Cases in GCC Countries

Only two GCC countries (Saudi Arabia and Oman) have recently reported indigenous transmission of malaria while the remaining countries have been malaria-free for the last several years. However, GCC countries continue to have imported malaria cases as they all have large expatriate populations [29,31,33,36,45,46,49]. Most of the expatriates in GCC countries originate from India, Pakistan, Afghanistan, Bangladesh and Philippines, located in South/Southeast Asia, and Sudan, Ethiopia, and Nigeria, located in sub-Saharan Africa. Table 3 presents number of local citizens and expatriate population (originating from malaria-endemic countries) among the six GCC countries. Expatriates from India, Pakistan, Afghanistan, Bangladesh, Philippines, Nigeria, Sudan and Ethiopia total more than 2 million individuals, which is more than the local Kuwaiti population of 1.43 million people (Table 3). Most of the countries from which expatriates originate are endemic or highly endemic for malaria [6,14]. Details regarding the incidence of malaria cases and malaria deaths reported by some countries during 2000–2019, and from where most of the expatriates working in GCC countries originate, are provided in Figure 3 and Figure 4. The data show that since 2010, the incidence of malaria cases has generally declined considerably in Ethiopia, India and Pakistan while it has shown an upward trend in Sudan (Figure 3). Similarly, the incidence of malaria deaths has also declined considerably since 2010 in Nigeria, Ethiopia and India while it has shown an upward trend in Sudan (Figure 4).

It has recently been suggested that continuous increases in long-distance travel and recent large migratory movements have changed the epidemiological characteristics of imported malaria in countries where malaria is not endemic, such as GCC countries [17]. Malaria infection is primarily imported to non-endemic countries by returning travelers from malaria-endemic countries. Many expatriates originating from malaria-endemic regions frequently (usually annually) travel to visit friends and relatives in their native countries. These individuals now make up the majority of malaria patients in non-endemic countries as they have partial immunity to malaria, resulting from repeated exposure to infection (Table 3) [29,31,33,36,45,46,49]. They also exhibit differences in the parasitological features, clinical manifestation and relatively lower odds for severe malaria compared to returning nationals of GCC countries, who have significantly higher odds for severe malaria because of their nonimmune status. Since the expatriate workers and their dependent family members contribute to a significantly high number of imported malaria cases in the GCC countries, a brief account of the current situation of malaria, including drug-resistant malaria, in these countries is also described here.

### 3.1. Characterization of Malaria Cases from India

India, with 1385 million people, has the second largest population (after China) in the world and Indians constitute the largest ethnic group among expatriate populations in nearly all GCC countries. India carries 4% of the global malaria burden and contributes nearly 87% of the total malaria cases in the WHO Southeast Asia Region. The latest WHO world malaria report has shown that nearly 5.6 million malaria cases occurred in India in 2019 [6]. However, the report also showed that India also contributed to the largest absolute reductions in malaria cases and deaths in the WHO Southeast Asia region from 2000 to 2019 (Figure 3 and Figure 4) [6]. The malaria outbreaks usually peak during summer and fall monsoon seasons which raise the overall number of malaria cases and deaths in the country [65,66].

Several studies have described the current situation of malaria in India. One study which employed a PCR-based assay reported an overall malaria prevalence of 19%, which varied from 6% in Oddanchatram, South India to 35% in Ratnagiri, West India [67]. Among malaria-positive patients, *P. falciparum* monoinfection was detected in 46% of the positive subjects. Furthermore, 38% subjects had *P. vivax* infection, 5% had *P. malariae* infection and 11% subjects had mixed infections with *P. falciparum* and *P. vivax* [67]. The study also showed that microscopy and rapid diagnostic tests had lower sensitivity than PCR-based assays, resulting in under diagnosis of malaria cases in the rural regions of India. One study carried out in the southwestern regions of India showed that *P. vivax* infections account for approximately 80% of malaria cases, which have been reported to cause severe malaria, leading to more deaths than *P. falciparum* infections [68]. Another recent study carried out in Mangaluru city in southwestern India examined a total of 579 malaria patients and reported that 364 (62.9%) had *P. vivax* infection, 150 (25.9%) had *P. falciparum* infection while 65 (11.2%) patients had mixed infection by two *Plasmodium* species [69]. The study also reported that the majority (506 or 87%) of malaria patients had mild malaria, which may be attributed to prompt treatment with antimalarial drugs or previous exposure to the malarial parasite. Although *P. vivax* was thought to dominate India in comparison to *P. falciparum*, a large recent study which examined 2333 blood samples collected from nine malaria-endemic Indian states by using multiple diagnostic tests (microscopy, rapid diagnostic tests and PCR assay) reported that the ratio of *P. vivax* to *P. falciparum* infection was almost the same (1.04:1) [70]. The study also reported that 13% of cases had *Plasmodium* spp. mixed infection.

Several studies have shown that mixed *Plasmodium* spp. infections are reported frequently from India [66,69,70,71]. Data from more recent studies have shown that *P. falciparum*, which had dominated India’s malaria cases previously, is now showing a decreasing trend over the past few years, from 65.4% in 2017 to 46.4% in 2019. India is a vast country with region-specific epidemiology of malaria. It has now become apparent that the majority of malaria cases in some regions in India are caused by *P. vivax*. On the contrary, the rising number of malaria infections in some other regions are caused mostly by *P. falciparum*, with only a minor contribution from *P. vivax*. Recent estimates show that India alone harbors nearly 47% of all *P. vivax* malaria cases worldwide, where 7 (Uttar Pradesh, Jharkhand, Chhattisgarh, West Bengal, Gujarat, Madhya Pradesh and Odisha) out of 36 Indian states account for nearly 90% of malaria cases [3].

In addition to the problems of *P. vivax* infection and mixed infections in some regions in India, increasing reports of drug resistance is another worrisome development [4]. There is also a higher risk of *P. vivax* parasitemia following treatment of *P. falciparum* malaria [72]. The epidemiology of drug-resistant malaria also varies considerably in different regions of India. One recent study has shown increasing prevalence of single nucleotide polymorphisms in the dihydrofolate reductase (*dhfr*) gene of *P. falciparum* from different regions in India [73]. Although chloroquine had been discontinued for nearly a decade, a higher abundance of mutated *P. falciparum* chloroquine-resistant transporter (*pfcrt*) and a lower prevalence of mutated *P. falciparum* multidrug resistance 1 (*pfmdr1*), which confer resistance to chloroquine, were observed [74,75]. Mutant alleles of *pvmdr1* gene were also observed among *P. vivax* from New Delhi Capital Region [76], Chandigarh in North India [77] and the southwestern coastal region in South India [78]. The presence of Kelch 13 mutations, which confer resistance to artemisinin-based combination therapy (such as artesunate/sulfadoxine/pyrimethamine), have also been reported from India [79]. Detection of artemisinin-resistant malaria parasites, some of which carry novel mutations, and increasing incidence of combination therapy failures are worrisome developments for malaria control in India and beyond.

Previous studies carried out in all GCC countries have shown that travelers or returning travelers from India have contributed significantly to imported malaria cases in Bahrain [29], Kuwait [44,45], Oman [31,32], Qatar [33,46,47], Saudi Arabia [49,50,51] and UAE [36] (Table 2).

### 3.2. Characterization of Malaria Cases from Pakistan

Pakistan is another highly malaria-endemic country in South Asia, reporting more than 670,000 cases of malaria and 3159 deaths in the year 2013 [14]. The latest WHO world malaria report has shown that 700,000 malaria cases occurred in Pakistan in 2019 [6]. A recent study in a highly malaria-endemic district in Pakistan examined blood smears of 2033 individuals who were suspected of malaria infection. The study utilized microscopy, a rapid diagnostic test (RDT) and a PCR-based assay and found 429 (21.1%) malaria-positive cases by at least one diagnostic test [80]. The data showed that the positivity by PCR-based assay, microscopy, and RDT was 30.5%, 17.7% and 16.4%, respectively. The study further showed that ~80% of malaria cases had *P. vivax* infection. Of the remaining cases, *P. falciparum* alone accounted for 11% of cases while *Plasmodium* spp. mixed infections were seen in 9% of malaria cases [80]. Another study had examined blood specimens from 216 febrile patients for the diagnosis of malaria in a tribe-governed region in Pakistan which also receives refugees from neighboring Afghanistan. The study found that most (86.5%) of the malaria cases were attributed to mono *P*. *vivax* infection. The authors showed that 11.8% patients had malaria due to *P*. *falciparum* and found that the ratio of *P. vivax* to *P. falciparum* infection has increased in provinces which border Afghanistan [81]. Another recent study which used RDTs to screen 31,041 individuals for malaria infection in three (Bannu, Dera Ismail Khan and Lakki Marwat) highly malaria-endemic districts of Khyber Pakhtunkhwa province in northern Pakistan reported an overall malaria prevalence of 13.8% in the population. The study further showed that the species prevalence of *P. vivax*, *P. falciparum* and *Plasmodium* spp. mixed infection were 92.4%, 4.7% and 2.9%, respectively [82]. These reports clearly highlight an increasing dominance of *P. vivax* in highly malaria-endemic regions of Pakistan.

Chloroquine-resistant *P. vivax* is now considered as an emerging threat in Pakistan. Mutations in the *pvmdr1* gene responsible for chloroquine resistance and *pvdhfr*, *pvdhps* genes associated with sulphadoxine/pyrimethamine resistance in *P. vivax* have been detected in recent studies from Pakistan [83,84]. Previous studies performed in GCC countries have shown that travelers or returning travelers from Pakistan have contributed to imported malaria cases in the GCC countries, as described in Table 2.

### 3.3. Characterization of Malaria Cases from Afghanistan

Afghanistan is another highly malaria-endemic country in Asia, where approximately half of the population is at risk of infection [84]. Previous studies have shown that the incidence of malaria in Afghanistan exceeded 200,000 cases in the year 2013, with 1783 deaths [14]. The latest WHO world malaria report has reported nearly 400,000 malaria cases in Afghanistan in 2019 and about 25% of these cases were due to *P. vivax* [6]. A recent study performed to determine malaria prevalence in Jalalabad city, employing real-time PCR with high resolution melting analysis on 296 blood samples, reported asymptomatic malaria in 26 (7.8%) individuals [85]. However, in this study 21 of 26 cases were due to *P. falciparum* while only 1 case of *P. vivax* was detected. The remaining 4 cases had mixed infection of *P. falciparum* and *P. vivax*. The study carried out in the southern region of Fars province in Iran also detected nearly all malaria cases among Afghan nationals, mostly including migrant workers, during 2006–2018 [86].

### 3.4. Characterization of Malaria Cases from Bangladesh

Bangladesh, another country with a population of more than 160 million people in the Indian subcontinent, is also a malaria-endemic country and nearly 17 million people are at risk of malaria infection [6]. In recent years, Bangladesh has been successfully accelerating its efforts to eliminate malaria from the country. As a result of concerted malaria control efforts, Bangladesh has seen a steady decline in the number of malaria cases during the period 2000–2019. According to the latest world malaria report, Bangladesh reported nearly 50,000 malaria cases in 2019 [6]. Although most malaria infections are caused by *P. falciparum*, the contribution of *P. vivax* malaria and antimalarial drug resistance have increased over the last 20 years [6,87]. Haque et al. [87] reviewed antimalarial drug resistance data from Bangladesh until June 2013 and reported that *P. falciparum* shows varying levels of resistance to chloroquine, mefloquine and sulfadoxine/pyrimethamine. A meta-analysis of data from malaria patients from three sites in Bangladesh has recently shown that 12–26% of *P. falciparum* malaria patients are at risk of *P. vivax* parasitemia after several weeks following treatment with antimalarial drugs [72]. Imported malaria cases have also been detected among returning Bangladeshi workers in GCC countries.

### 3.5. Characterization of Malaria Cases from the Philippines

The Philippines, located in the WHO Western Pacific Region, is also a malaria-endemic country. Philippines reported more than 550,000 malaria cases and 230 deaths in the year 2013 [14]. However, the number of malaria cases and deaths have declined considerably from the year 2010 to 2018. The two regions with the highest prevalence of malaria in the Philippines are Palawan province and Mindanao Islands [3,6]. According to the latest world malaria report, the Philippines reported nearly 40,000 malaria cases in 2019 [6]. Zoonotic malaria infections have also been reported from the Philippines in recent years. Some studies have shown mixed *P. falciparum/P. vivax* malaria infections as well as zoonotic malaria due to *P. knowlesi* in younger individuals who were working in agricultural fields within Palawan province in the Philippines [88,89]. Drug-resistant malaria cases have also emerged as mutations in dihydropteroate synthase (pvdhps) and dihydrofolate reductase (pvdhfr), genes associated with sulfadoxine/pyrimethamine drug resistance, have been reported among *P. vivax* isolates in Palawan province in the Philippines [90]. Imported malaria cases have been detected among returning Filipino workers in GCC countries [45,46].

### 3.6. Characterization of Malaria Cases from Nigeria

Nigeria, with a population of 208 million individuals and located in the sub-Saharan region, is the most populous country in Africa. Nigeria is the leading contributor of malaria cases and deaths not only in the WHO African Region but also globally. According to the latest WHO world malaria report, Nigeria reported nearly 60 million malaria cases (corresponding to 27% of total malaria cases) and 23% of total malaria deaths worldwide in 2019 [6]. Contrary to the malaria trends seen in most of the malaria-endemic countries of the world, the number of malaria cases as well as malaria-related deaths have increased in 2019 compared to 2018 in Nigeria [6]. Nigeria is also one of eleven countries which have reported Pfhrp2/3 deletions in the *P. falciparum* parasite, which compromise the utility of HPR2-based rapid diagnostic tests for the detection of malaria cases [6]. One study showed that the levels of *P. falciparum* resistance to sulfadoxine/pyrimethamine have remained low from 2000 to 2020 in western Africa, including in Nigeria [91]. However, the rising incidence of polymorphisms in *P. falciparum* genes (Pfk13, Pfmdr1, PfATPase6 and Pfcrt) associated with drug resistance to first-line drugs (artemisinin combination therapy) is a major challenge in malaria control efforts [92]. Previous studies from GCC countries have shown that travelers or returning travelers from Nigeria have presented with imported malaria cases in Kuwait, Qatar, Saudi Arabia and UAE [36,44,45,46,47,49,51].

### 3.7. Characterization of Malaria Cases from Ehiopia

Ethiopia, a land-locked country located in the Horn of Africa, is also a significant contributor for total malaria cases in the WHO African Region. According to the latest WHO world malaria report, Ethiopia reported nearly 2.5 million malaria cases in 2019 [6]. Although the number of total malaria infections in Ethiopia is still high, the country has shown considerable progress in reducing the total burden of malaria over the past two decades by employing widespread use of LLINs and IRS to protect individuals from mosquito bites [93]. In recent years, Ethiopia has shifted efforts from the control of malaria to elimination of malaria in some regions of the country. The data reported in a recent study carried out in the Harari Region have shown that malaria incidence has declined from 42.9 cases per 1000 population in 2013 to only 6.7 cases per 1000 population in 2019 [93]. Furthermore, malaria-related deaths also decreased from 4.7 deaths per 1,000,000 persons annually in 2013 to zero in 2015. The data also showed that *P. falciparum, P. vivax* and mixed infections accounted for 69.2%, 30.6% and 0.2% of all malaria cases, respectively, in the Harari Region. Ethiopia is now determined to eliminate malaria by 2030. In this regard, the country is actively pursuing artemether/lumefantrin treatment as one of the cornerstone strategies for uncomplicated *P. falciparum* malaria control [94]. Previous studies from GCC countries have shown that travelers or returning travelers from Ethiopia have contributed to imported malaria cases in all GCC countries [32,36,45,46,47,49,51].

### 3.8. Characterization of Malaria Cases from Sudan

Sudan, a sub-Saharan African country, belongs to the WHO Eastern Mediterranean Region. Sudan reported nearly 2.4 million malaria cases in 2019, the highest number reported by any country in this WHO region [5]. Recent reports of the development of drug resistance in the malarial parasite is a major threat to malaria control programs in Sudan. One recent study based on molecular makers of antimalarial drug resistance in *P. falciparum* isolates was carried out during 2015–2017 [95]. The authors reported failure of artemisinin-based combination therapy for the treatment of uncomplicated malaria in Sudan. This was attributed to the high prevalence of mutations in *P. falciparum* drug resistance genes. In another study, the authors reported evolution of drug resistance in *P. falciparum* following artemisinin combination therapy [96]. The study reported high frequency of mutations in *Pfcrt*, *Pfdhfr* and *Pfdhps*, which are associated with chloroquine and sulfadoxine/pyrimethamine (SP) resistance in *P. falciparum*. Travelers or returning travelers from Sudan have contributed to the number of imported malaria cases in all GCC countries [32,36,45,46,47,49,51].

## 4. Prospects for Malaria Elimination in GCC Countries

As stated above, four (Bahrain, Kuwait, Qatar and UAE) of six GCC countries have been malaria-free for several decades, while Oman has reported indigenous transmission of malaria cases only rarely in recent years [24,28,29,30,31,36,44,45,46,47,53,55,56]. Saudi Arabia is the only GCC country which continues to have indigenous transmission of malaria cases, even though their numbers have declined considerably in recent years [35,48,49,50,51,58,59,60]. However, travel-related imported malaria cases continue to occur in all GCC countries as they have large expatriate populations originating from malaria-endemic countries [24,28,29,30,31,35,36,44,45,46,47,48,49,50,51,55,56,58,59,60]. More stringent border controls due to COVID-19 in the last several months have likely reduced the incidence of imported malaria cases in nearly all GCC countries [Jamshaid Iqbal, unpublished observations]. The GCC countries have also introduced more stringent measures to reduce the likelihood of importing malaria cases by prior screening of newly arrived expatriates in their native countries to detect malaria infection prior to their departure to GCC countries [45,47,49,59]. As of now, no mosquito species are known to breed in Bahrain, Kuwait and the United Arab Emirates, and vigilance is being exercised by relevant departments of these governments to maintain this status [29,36,45]. Furthermore, efficient mosquito vector control efforts have either reduced or eliminated breeding foci in Oman, Qatar and Saudi Arabia with the result that indigenous transmission of infections has either not been reported (Qatar) [33,46,47,55,56], is reported very infrequently (Oman) [30,31,32] or has reduced considerably (Saudi Arabia) [34,35,48,49,50,51,58,59,60]. However, a few problem areas have also remained or emerged recently. These include widespread insecticide resistance in *An. stephensi* that is prevalent in the WHO Eastern Mediterranean Region, including countries bordering or in the near vicinity of GCC countries [97], and temperature-dependent expansion of the range of mosquito species into new geographic areas and their interaction with resident mosquito species (e.g., expansion of *An. stephensi* in the Middle East and North Africa where *Ae. aegypti* is endemic) [98].

## 5. Conclusions

Although the highest number of malaria cases and deaths occur in sub-Saharan African and South/Southeast Asian countries, imported cases of malaria occur in all GCC countries, though indigenous transmission of malaria has been reported in only two (Oman and Saudi Arabia) countries. Oman has been mainly malaria-free in the last few years and Saudi Arabia is also striving for malaria elimination. However, imported malaria is being increasingly recognized as a new public health challenge and its control is complicated, as GCC countries have a large and dynamic expatriate population originating from many malaria-endemic countries. Drug-resistant *P. vivax* strains have emerged and transmission of such infections or relapse through imported malaria cases in malaria-free GCC countries have further complicated disease management in returning travelers.

Other possible risk factors for local transmission of infection have recently emerged in GCC or other nearby countries, which include a change in the present ecological and climate conditions due to an enthusiastic drive of planting to make these countries green. These conditions may raise larval density levels of the mosquito vector which may support local transmission of infection. Temperature-dependent expansion of the range of mosquito species into new geographic areas and their interaction with resident mosquito species has also been noted in recent years. Some countries have reported failure of artemisinin-based combination therapies for the treatment of uncomplicated malaria and mutations in *P. falciparum* drug resistance genes have been detected. High frequency of mutations in *Pfcrt*, *Pvmdr1, Pfdhfr, pvdhfr*, *Pfdhps* and *pvdhps* have also been detected which are associated with chloroquine and sulfadoxine/pyrimethamine (SP) resistance in *P. falciparum* and *P. vivax* in some countries. Thus, for many of the GCC countries to remain malaria free, early diagnosis, appropriate treatment of the disease and regular monitoring for drug resistance in *Plasmodium* spp. need to be uninterruptedly maintained.

## Figures and Tables

**Figure 1 microorganisms-09-01431-f001:**
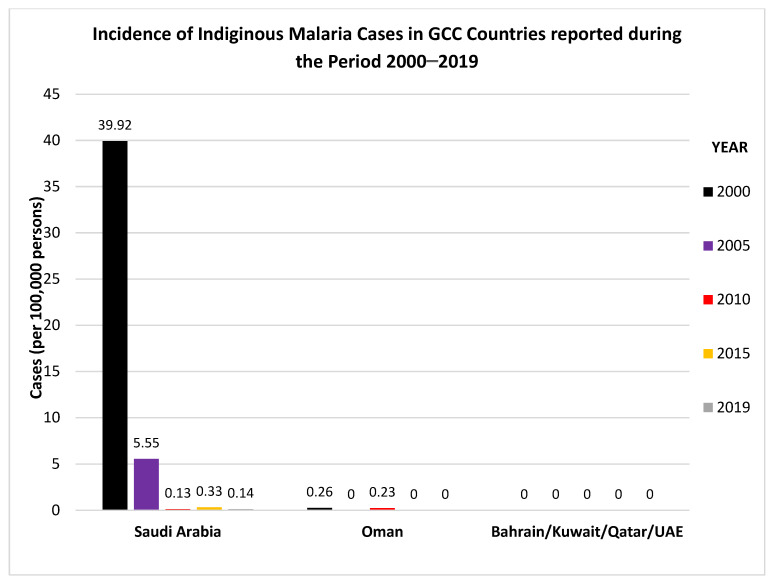
Incidence of indigenous malaria cases in GCC Countries reported during the Period 2000–2019. The graphs were made using data adopted from WHO, World Malaria Reports, 2019 and 2020 and Malaria Atlas Project 2019 (https://malariaatlas.org/malaria-burden-data-download/ accessed on 20 March 2021).

**Figure 2 microorganisms-09-01431-f002:**
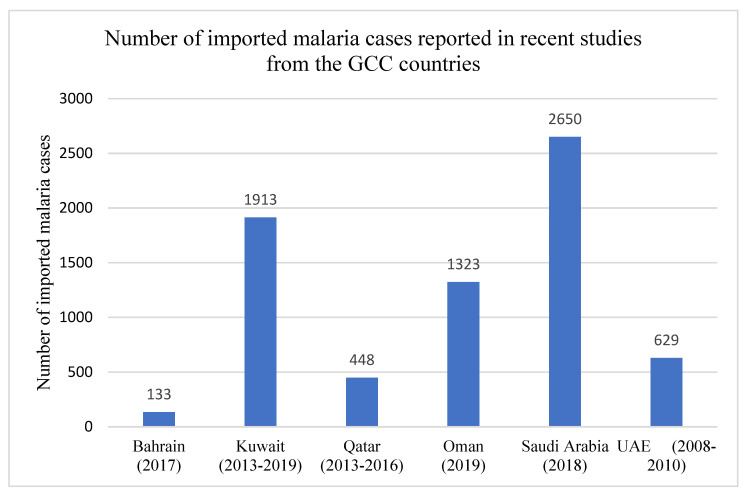
Number of imported malaria cases in GCC countries. The graphs were made using data adopted from WHO, World Malaria Reports, 2019 and 2020 and the Malaria Atlas Project 2019.

**Figure 3 microorganisms-09-01431-f003:**
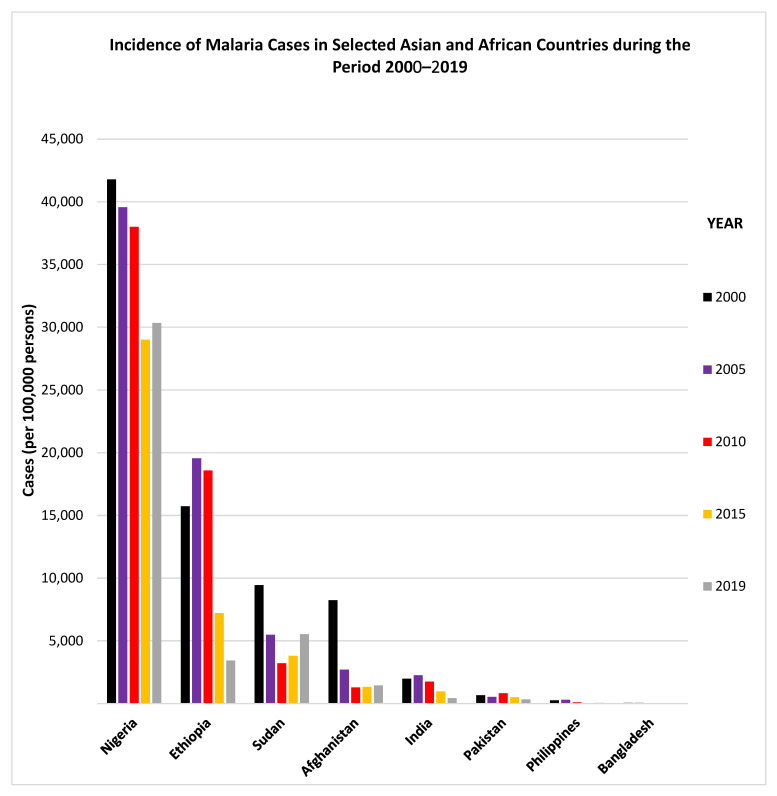
Incidence of malaria cases in selected Asian and African countries during the period 2000–2019. The graphs were made using data adopted from WHO, World Malaria Reports, 2019 and 2020 and the Malaria Atlas Project 2019.

**Figure 4 microorganisms-09-01431-f004:**
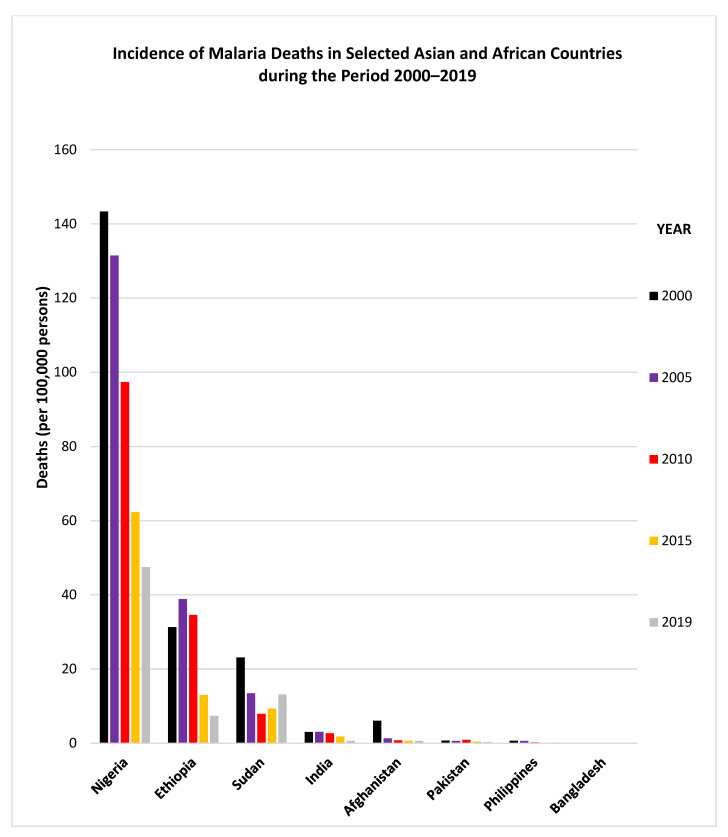
Incidence of malaria deaths in selected Asian and African countries during the period 2000–2019. The graphs were made using data adopted from WHO, World Malaria Reports, 2019 and 2020 and the Malaria Atlas Project 2019.

**Table 1 microorganisms-09-01431-t001:** An updated list of major malaria vectors and indigenous malaria status in GCC countries.

Country	Major Vectors	Indigenous Malaria Status *	References Number
Bahrain	None	Malaria-free since 2012	Mahmood RA, 1992 [28] Ismaeel et al., 2004 [29]
Kuwait	None ^#^	Malaria-free since 1963	
Oman	*Anopheles culicifacies*, very low density	very low and unstable (3 indigenous malaria cases in 2016)	Hassan KS, 2017 [30] Simon et al., 2017 [31] MOH, Annual Report 2019 [32]
Qatar	*An. stephensi, An. multicolor*	Malaria-free since 2012	Al-Kuwari MG, 2009 [33]
Saudi Arabia	*An. Stephensi* **, *An. sergenti*, *An. gambiae*, *An. arabiensis*, *An. bwambae*, *An. coluzzii*, *An. merus*	Low and unstable (mainly in the southeastern region) 61 cases (57 *P. falciparum* and 4 *P. vivax*) in 2018	Alahmed, AM et al., 2019 [34] Hawash Y et al., 2019 [35]
United Arab Emirates	None	Malaria-free since 2007	Nilles et al., 2014 [36]

* Official declaration by the WHO [24]; ** main malaria vector in the Eastern Region of Saudi Arabia. ^#^
*Anopheles stephensi* and *An. sergenti* are the main vectors in the neighboring country, Iraq, which may cross over to Kuwait.

**Table 2 microorganisms-09-01431-t002:** Indigenous and imported malaria cases reported in different recent studies from the GCC countries.

Country	Duration of Study	No. of Indigenous and Imported Malaria Cases	No. of Imported Malaria Cases Detected among Nationals (Citizens) or Expatriates									Malaria Cases by *Plasmodium* spp.				Reference
			Citizens	India	Pakistan	AFGN	BGD	Nigeria	Sudan	Other African Countries	Others	*P. falciparum*	*P. vivax*	Pf/Pv Mixed	Others	
Bahrain	1992–2001	0 and 1572	N. A.	629	566	N. A.	31	NA	63	N. A.	283	220	1346	5	1	Ismaeel et al., 2004 [29]
Bahrain	2017	0 and 133	N.A.	N.A.	N.A.	N.A.	N.A.	N.A.	N.A.	N.A.	N.A.					WHO/EMRO Annual Report 2017 [37]
Kuwait	1985–2000	0 and 6776 *	39	3569	1057	1871	0	89	48	N. A.	133	1137	5207	395	0	Iqbal et al., 2003 [44]
Kuwait	2013–2018	0 and 1913	18	1012	390	94	5	16	48	275	55	361	124	1383	45	Iqbal et al., 2020 [45]
Oman	2014	53 and 1	1	14	6	0	32	0	0	0	1	0	54	0	0	Simon et al., 2017 [31]
Oman	2019	15 and 1323	N. A.	N. A.	N. A.	N. A.	N. A.	N. A.	N. A.	N. A.	N. A.	1080	206	0	52	MoH, Oman [32]
Qatar	2004–2006	0 and 438	N. A.	210	128	N. A.	N. A.	N. A.	37	N. A.	63	60	175	0	203	Al-Kuwari, 2009 [33]
Qatar ^a^	2008–2015	0 and 4092	14	812	772	0	0	200	0	0	14	404	2336 ^b^	0	229 ^b^	Farag et al. 2018 [46]
Qatar	2013–2016	0 and 448	1	168	108	0	0	16	74	8	73	118	318	12	0	Al-Rumhi et al., 2020 [47]
Saudi Arabia	2000–2014	5522 and 9930	N. A.	N. A.	N. A.	N. A.	N. A.	N. A.	N. A.	N. A.	N. A.	N. A.	N. A.	N. A.	N. A.	El Hassan et al., 2015 [48]
Saudi Arabia	2008–2011	318 ^c^	16	37	108	5	3	53	12	7	77	204	103	0	11	Memish et al. 2014 [49]
Saudi Arabia	2012–2015	121 and 224	113	28	73	3	0	0	48	37	43	212	128	2	3	Alshahrani et al., 2016 [50]
Saudi Arabia	2016	4 and 22	4	4	5	0	0	2	10	1	0	13	13	0	0	Soliman et al. 2018 [51]
Saudi Arabia	2016–2018	5 and 25	11	2	2	0	0	0	4	0	11	23	5	2	0	Hawash et al., 2019 [35]
Saudi Arabia	2018	61 and 2650	N. A.	N. A.	N. A.	N. A.	N. A.	N. A.	N. A.	N. A.	N. A.	57	4	0	0	WHO Malaria Report 2019 [3]
UAE	2008–2010	0 and 629	0	338	228	0	0	14 **	14 **	14 **	21	122	493	14	0	Nilles et al., 2014 [36]

* includes only the post-war period of the study i.e., 1992–1997; ^a^ patient nationality data is provided for 1816 cases detected during 2013–2015 only; ^b^ species distribution available for 2969 cases detected in 2008–2009 and 2012–2015; ^c^ nearly all cases were imported malaria cases but exact numbers are not available; ** 42 cases were reported from sub-Saharan Africa in the cited reference and were equally divided among the three countries listed here. Other African countries: all sub-Saharan countries, Ghana, Ethiopia and Sierra Leone. Others: other countries including GCC, Indonesia, Sri Lanka and Nepal; UAE., United Arab Emirates; N. A., Not available. It is apparent from these studies that yearly occurrence of imported malaria cases is variable among GCC countries [29,31,32,35,36,44,45,46,47,48,49,50,51]. These variations are expected as they have varying proportion of expatriates from malaria-endemic countries. Although the data are not available for the same year, the largest number of imported malaria cases per year were detected in Saudi Arabia [48] while the smallest number were detected in Bahrain [37] and Qatar [33] which generally reflect the number of expatriates in these countries from malaria-endemic countries. Thus, a continuous influx of imported malaria cases via expatriate workers originating from malaria-endemic countries of Africa, the Indian subcontinent and Southeast Asia sustains the threat of local transmission of infection in GCC countries.

**Table 3 microorganisms-09-01431-t003:** Number of local citizens and expatriate population (originating from malaria-endemic countries) among the GCC countries.

GCC	Total Population in	Population of Citizens	Population of Expatriates from Different Countries in GCC Countries in 2019							
Country	2019 (in Millions)	in 2019 (in Millions)	India	Pakistan	Afghanistan	Bangladesh	Philippines	Nigeria	Sudan	Ethiopia
Bahrain	1.59	1.21	318,547	78,638	690	82,518	50,585	2054	7917	713
Kuwait	4.77	1.43	1,124,256	330,824	2826	370,844	192,143	4702	48,204	3806
Qatar	2.68	0.33	698,088	235,876	1602	263,086	168,461	4152	23,954	1700
Oman	4.7	2.6	1,325,444	240,965	N. A.	304,917	44,546	N. A.	19,155	N. A.
Saudi Arabia	34	23.2	2,440,489	1,447,071	469,324	1,246,052	628,894	N. A.	469,324	160,192
UAE	9.7	1.2	3,419,875	981,536	8071	1,079,013	556,407	15,465	131,254	10,886

UAE: United Arab Emirates; N.A. not available.

## Data Availability

Data sharing not applicable.

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
