# Peer review of "Current Epidemiological Characteristics of Imported Malaria, Vector Control Status and Malaria Elimination Prospects in the Gulf Cooperation Council (GCC) Countries"

_microorganisms, 2021, doi:10.3390/microorganisms9071431_

Round 1

Reviewer 1 Report

This is a very interesting paper describing the efforts of the Gulf Cooperation Council (GCC) countries in malaria elimination. The paper offers a comprehensive view of the past and present strategies for malaria elimination, and the challenges of imported malaria in countries with strong connection to highly malaria endemic settings. The manuscript is well written, and I would only have a few minor comments.

  • There are a lot of repetitions between the different sections and paragraphs, especially about malaria burden worldwide. I would advise the authors to reduce the text by removing those repetitions. For example the first two paragraphs of section 2: "Brief account of current malaria status in major world regions"are repetitions from the introduction section. Please review the ext, there are many more repetitions.
  • Please review your references, I've picked some examples below with false statements and reference not supporting the statements.
  • In section 2, you mention that PfKelch 13 mutations have been detected in many countries and area associated with partial artemisinin resistance. However you should mention, that only a few number of these mutations are associated with artemisinin resistance, and many other not.
  • In section 3.2.4. In this section you cite reference 47, and mention that novel PfKelch 13 mutations have been identified. This is misleading, those mutations are not associated with artemisinin resistance, and not mutation associated with artemisinin resistance has been reported in this publication.
  • Section 4. Figures 3 and 4 do report incidence, and not absolute numbers of cases and deaths
  • Section 4. You mention that expatriates from malaria endemic settings experience a higher prevalence of severe malaria and complications compared to non-immune travelers, despite their acquired immunity. How do you explain that? Could be that due to socio-economic status and access to healthcare?
  • Section 4.1. You cite reference 78 to describe PfKelch 13 mutations associated with artemisinin resistance. However this publication is only reporting pvcrt and pvmdr1 mutations associate with chloroquine in vivax.
  • Section 4.2. Same here, reference 82 does not contain any data on molecular markers of resistance
  • Section 4.8. Again reference 94 does not contain resistance markers data and data are not from Sudan, but from Ethiopia
  • Section 6: In your conclusion, you mention that many countries have reported treatment failures with ACTs, actually this is not correct, only a few countries in the Sub Mekong region have reported high treatment failure rates with ACTs, and there are few reports of moderate treatment failure rate outside that region.

Author Response

Reviewer 1

Reviewer comments:

This is a very interesting paper describing the efforts of the Gulf Cooperation Council (GCC) countries in malaria elimination. The paper offers a comprehensive view of the past and present strategies for malaria elimination, and the challenges of imported malaria in countries with strong connection to highly malaria endemic settings. The manuscript is well written, and I would only have a few minor comments.

Authors response: We thank the reviewer for the positive comments. No specific comments to respond to.

Reviewer comments:

  • There are a lot of repetitions between the different sections and paragraphs, especially about malaria burden worldwide. I would advise the authors to reduce the text by removing those repetitions. For example the first two paragraphs of section 2: "Brief account of current malaria status in major world regions"are repetitions from the introduction section. Please review the ext, there are many more repetitions.

Authors response: The repetitive text has been deleted, as suggested by the reviewer.

Reviewer comments:

  • Please review your references, I've picked some examples below with false statements and reference not supporting the statements.

Authors response: This was caused by omission of citation of an additional reference (Reference no. 75) on Page 13, Para 2, Line 13. This is the reason that the few references did not match with the text from this point onward. We regret the mistake which has now been corrected.

Reviewer comments:

  • In section 2, you mention that PfKelch 13 mutations have been detected in many countries and area associated with partial artemisinin resistance. However, you should mention, that only a few number of these mutations are associated with artemisinin resistance, and many other not.

Authors response: The text has been modified, as suggested by the reviewer.

Reviewer comments:

  • In section 3.2.4. In this section you cite reference 47, and mention that novel PfKelch 13 mutations have been identified. This is misleading, those mutations are not associated with artemisinin resistance, and not mutation associated with artemisinin resistance has been reported in this publication.

Authors response: We thank the reviewer for this comment. The text has now been modified.

Reviewer comments:

  • Section 4. Figures 3 and 4 do report incidence, and not absolute numbers of cases and deaths

Authors response: The text has been modified, as suggested by the reviewer.

Reviewer comments:

  • Section 4. You mention that expatriates from malaria endemic settings experience a higher prevalence of severe malaria and complications compared to non-immune travelers, despite their acquired immunity. How do you explain that? Could be that due to socio-economic status and access to healthcare?

Authors response: We thank the reviewer for this important comment. The text has now been clarified/modified under section 4 (section 3 in the revised manuscript), last paragraph.

Reviewer comments:

  • Section 4.1. You cite reference 78 to describe PfKelch 13 mutations associated with artemisinin resistance. However this publication is only reporting pvcrt and pvmdr1 mutations associate with chloroquine in vivax.

Authors response: As explained above, this should have been reference no. 79. We regret the mistake which has now been corrected.

Reviewer comments:

  • Section 4.2. Same here, reference 82 does not contain any data on molecular markers of resistance

Authors response: As explained above, this should have been reference no. 83. We regret the mistake which has now been corrected.

Reviewer comments:

  • Section 4.8. Again reference 94 does not contain resistance markers data and data are not from Sudan, but from Ethiopia

Authors response: As explained above, this should have been reference no. 95. We regret the mistake which has now been corrected.

Reviewer comments:

  • Section 6: In your conclusion, you mention that many countries have reported treatment failures with ACTs, actually this is not correct, only a few countries in the Sub Mekong region have reported high treatment failure rates with ACTs, and there are few reports of moderate treatment failure rate outside that region.

Authors response: The text has been modified, as suggested by the reviewer.

Reviewer 2 Report

The authors present a review of studies related to the epidemiology of malaria in the GCC countries.

The review is  in a form of a listing of some (all?) studies with a short description of  them. Now evaluation or classification of the quality of the studies is made. It is a narrative description of studies and their results. No scientific systematic is identifiable in the review. It is mainly based on the view of the authors which study should be described or not

Overall a simple descriptive presentation of a series of studies supported by some graphs.

The abstract is more of a reduced introduction and should be rewritten.The abstract should contain information about background, methods, results and discussion. Even if it is a review article. Background is presented but information on methods, results and discussion is missing.

Section 1 and 2 could be substantially reduced. The authors can go into the specific issue they address faster. There is no need to be so detailed about the malaria epidemiology in the world.

Figure 2 is not particularly informative since it refers to different periods for the individuals countries. It gives some overview but what is its message of this table? Is the number among the individual countries variable. How variable? Is it standardised to show it? A better elaboration of the table could be more informative.

There is no information about the methodology the authors used. How did they select the studies they used for their review? What type of studies they included/excluded? More information on methodology is needed.

Author Response

Reviewer 2

Reviewer comments:

The authors present a review of studies related to the epidemiology of malaria in the GCC countries.

The review is  in a form of a listing of some (all?) studies with a short description of  them. Now evaluation or classification of the quality of the studies is made. It is a narrative description of studies and their results. No scientific systematic is identifiable in the review. It is mainly based on the view of the authors which study should be described or not

Authors response: We thank the reviewer for the positive comments. All studies indexed in PubMed describing epidemiological characteristics of indigenous and imported malaria cases, vector control status and how malaria infections can be controlled to achieve malaria elimination in GCC countries were reviewed and discussed in the manuscript.

Reviewer comments:

Overall a simple descriptive presentation of a series of studies supported by some graphs.

Authors response: We thank the reviewer for the positive comments. No specific comments to respond to.

Reviewer comments:

The abstract is more of a reduced introduction and should be rewritten.The abstract should contain information about background, methods, results and discussion. Even if it is a review article. Background is presented but information on methods, results and discussion is missing.

Authors response: The Abstract has been re-written, as suggested by the reviewer.

Reviewer comments:

Section 1 and 2 could be substantially reduced. The authors can go into the specific issue they address faster. There is no need to be so detailed about the malaria epidemiology in the world.

Authors response: Sections 1 and 2 have been combined and substantially reduced, as suggested by the reviewer.

Reviewer comments:

Figure 2 is not particularly informative since it refers to different periods for the individual countries. It gives some overview but what is its message of this table? Is the number among the individual countries variable. How variable? Is it standardised to show it? A better elaboration of the table could be more informative.

Authors response: The data in Table 2 shows that the number of imported malaria cases in different studies and the Plasmodium species causing malaria infections vary considerably even from the same location due to the large and highly dynamic expatriate population in the GCC countries. This aspect has now been added in the revised manuscript. In addition, the data in Table 2 also highlights a gradual decrease in the number of indigenous (Oman & Saudi Arabia) over the years, as a consequence to malaria control/preventive strategies adopted by the countries.    

Reviewer comments:

There is no information about the methodology the authors used. How did they select the studies they used for their review? What type of studies they included/excluded? More information on methodology is needed.

Authors response: As stated above, all studies indexed in PubMed describing epidemiological characteristics of indigenous and imported malaria cases, vector control status and how malaria infections can be controlled to achieve malaria elimination in GCC countries were included. This information has now been added in the revised manuscript, as suggested by the reviewer.

Round 2

Reviewer 2 Report

The authors responded to my comments and addressed them in the manuscript.